# FAIR DIFFERENTIAL PRIVACY CAN MITIGATE THE DISPARATE IMPACT ON MODEL ACCURACY

## ABSTRACT

The techniques based on the theory of differential privacy (DP) has become a standard building block in the machine learning community. DP training mechanisms offer strong guarantees that an adversary cannot determine with high confidence about the training data based on analyzing the released model, let alone any details of the instances. However, DP may disproportionately affect the underrepresented and relatively complicated classes. That is, the reduction in utility is unequal for each class. This paper proposes a fair differential privacy algorithm (FairDP) to mitigate the disparate impact on each class's model accuracy. We cast the learning procedure as a bilevel programming problem, which integrates differential privacy with fairness. FairDP establishes a self-adaptive DP mechanism and dynamically adjusts instance influence in each class depending on the theoretical bias-variance bound. Our experimental evaluation shows the effectiveness of FairDP in mitigating the disparate impact on model accuracy among the classes on several benchmark datasets and scenarios ranging from text to vision.

## 1 INTRODUCTION

Protecting data privacy is a significant concern in many data-driven decision-making applications (Zhu et al., 2017), such as social networking service, recommender system, location-based service. For example, the United States Census Bureau will firstly employ differential privacy to the 2020 census data (Bureau, 2020). Differential privacy (DP) guarantees that the released model cannot be exploited by attackers to derive whether one particular instance is present or absent in the training dataset (Dwork et al., 2006). However, DP intentionally restricts the instance influence and introduces noise into the learning procedure. When we enforce DP to a model, DP may amplify the discriminative effect towards the underrepresented and relatively complicated classes (Bagdasaryan et al., 2019; Du et al., 2020; Jaiswal & Provost, 2020). That is, reduction in accuracy from non-private learning to private learning may be uneven for each class. There are several empirical studies on utility reduction: (Bagdasaryan et al., 2019; Du et al., 2020) show that the model accuracy in private learning tends to decrease more on classes that already have lower accuracy in non-private learning. (Jaiswal & Provost, 2020) shows different observations that the inequality in accuracy is not consistent for classes across multiple setups and datasets. It needs to be cautionary that although private learning improves individual participants' security, the model performance should not harm one class more than others.

The machine learning model, specifically in supervised learning tasks, outputs a hypothesis $f(\boldsymbol{x}; \theta)$ parameterized by $\theta$, which predicts the label $y$ given the unprotected attributes $\boldsymbol{x}$. Each instance's label $y$ belongs to a class $k$. The model aims to minimize the objective (loss) function $\mathcal{L}(\theta; \boldsymbol{x}, y)$, i.e.,

$$\theta^* := \arg\min_{\theta} \ \mathbb{E}\left[\mathcal{L}(\theta; \boldsymbol{x}, y)\right]. \tag{1}$$

Our work builds on a recent advance in machine learning models' training that uses the differentially private mechanism, i.e., DPSGD (Abadi et al., 2016) for releasing model. The key idea can be extended to other DP mechanisms with the specialized noise form (generally Laplacian or Gaussian distribution). The iterative update scheme of DPSGD at the $(t+1)$-th iteration is of the form

$$\tilde{\theta}^{t+1} = \tilde{\theta}^t - \mu^t \cdot \frac{1}{n} \left( \sum_{i \in \mathcal{S}^t} \frac{g^t(\boldsymbol{x}_i)}{\max(1, \frac{\|g^t(\boldsymbol{x}_i)\|_2}{C})} + \xi \mathbf{1} \right), \tag{2}$$

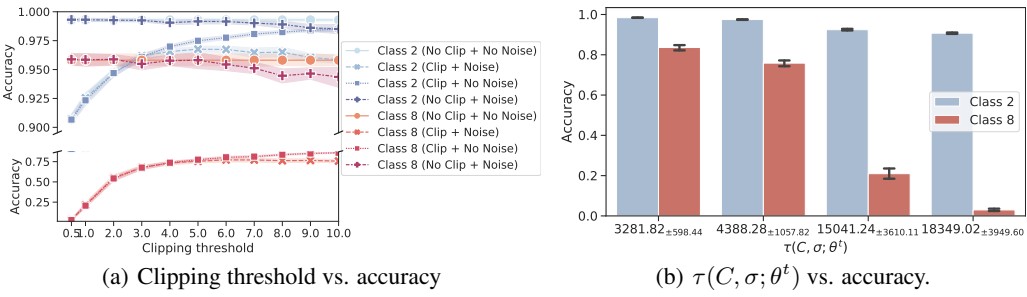

(a) Clipping threshold vs. accuracy        (b) $\tau(C, \sigma; \theta^t)$ vs. accuracy.

Figure 1: Effect of clipping and noise in differentially private mechanism and $\tau(C, \sigma; \theta^t)$ on MNIST dataset

where $n$ and $\mu^t$ denote the batch size and step-size (learning rate) respectively; $\mathcal{S}^t$ denotes the randomly chosen instance set; the vector $\mathbf{1}$ denotes the vector filled with scalar value one; and $g^t(\boldsymbol{x}_i)$ denotes the gradient of the loss function in (1) at iteration $t$, i.e., $\nabla\mathcal{L}(y_i; \theta^t, \boldsymbol{x}_i)$. The two key operations of DPSGD are: i) clipping each gradient $g^t(\boldsymbol{x}_i)$ in $\ell_2$-norm based on the threshold parameter $C$; ii) adding noise $\xi$ drawn from Gaussian distribution $\mathcal{N}(0, \sigma^2 C^2)$ with a variance of noise scale $\sigma$ and the clipping threshold parameter $C$. These operations enable training machine learning models with non-convex objectives at a manageable privacy cost. Based on the result of traditional SGD, we theoretically analyze the sufficient decrease type scheme of DPSGD, i.e.,

$$\mathbb{E}\left[f(\theta^{t+1})\right] \leqslant f(\theta^t) + \mathbb{E}\left[\langle\nabla f(\theta^t), \theta^{t+1} - \theta^t\rangle\right] + \frac{L}{2}\mathbb{E}\left[\left\|\theta^{t+1} - \theta^t\right\|^2\right] + \tau(C, \sigma; \theta^t), \quad (3)$$

where the last term $\tau(C, \sigma; \theta^t)$ denotes the gap of loss expectation compared with ideal SGD at this $(t+1)$-th iteration, and related with parameters $C$, and $\sigma$. The term $\tau(C, \sigma; \theta)$, which can be called *bias-variance* term, can be calculated mathematically as

$$\underbrace{2(1 + \frac{1}{\mu^t L})\|\nabla f(\theta)\| \cdot \eta + \eta^2}_{\text{Clipping bias}} + \underbrace{\frac{1}{n^2} \cdot \sigma^2 C^2 |\mathbf{1}|}_{\text{Noise variance}}, \quad (4)$$

where $L$ denotes the Lipschitz constant of $f$; $|\mathbf{1}|$ denotes the vector dimension; and we have

$$\eta := \frac{1}{n}\sum_{\mathbb{I}_{\|g^t(\boldsymbol{x}_i)\| > C}}\left(\|g^t(\boldsymbol{x}_i)\| - C\right),$$

where $\mathbb{I}_{\|g^t(\boldsymbol{x}_i)\| > C}$ denotes the cardinality number of satisfying $\|g^t(\boldsymbol{x}_i)\| > C$. The detailed proof of (3) and (4) can be found in Appendix A. $\tau(C, \sigma)$ is consist of *clipping bias* and *noise variance* terms, which means the amount that the private gradient differs from the non-private gradient due to the influence truncation and depending on the scale of the noise respectively. As a result, we call $\tau(C, \sigma)$ the bias-variance term.

As underrepresented class instances or complicated instances manifest differently from common instances, a uniform threshold parameter $C$ may incur significant accuracy disparate for different classes. In Figure 1(a), we employ DPSGD(Abadi et al., 2016) on the unbalanced MNIST dataset (Bagdasaryan et al., 2019) to numerical study the inequality of utility loss (i.e., the prediction accuracy gap between private model and non-private model) caused by differential privacy. For the unbalanced MNIST dataset, the underrepresented class (Class 8) has significantly larger utility loss than the other classes (e.g., Class 2) in the private model. DPSGD results in a 6.74% decrease in accuracy on the well-represented classes, but accuracy on the underrepresented class drops 74.16%. Training with more epochs does not reduce this gap while exhausting the privacy budget. DPSGD obviously introduces negative discrimination against the underrepresented class (which already has lower accuracy in the non-private SGD model). Further, Figure 1(b) shows the classification accuracy of different sub-classes for $\tau(C, \sigma; \theta)$ on the unbalanced MNIST dataset. Larger bias-variance term $\tau(C, \sigma; \theta)$ (determined by $C$ and $\sigma$) results in more serious accuracy bias on different classes, while similar results are also shown in (Bagdasaryan et al., 2019; Du et al., 2020; Jaiswal & Provost,

2020). Both theoretical analysis and experimental discussion suggest that minimizing the clipping bias and noise variance simultaneously could learn "better" DP parameters, which mitigates the accuracy bias between different classes. This motivates us to pursue fairness with a self-adaptive differentially privacy scheme[1].

This paper proposes a fair differential privacy algorithm (FairDP) to mitigate the disparate impact problem. FairDP introduces a self-adaptive DP mechanism and automatically adjusts instance influence in each class. The main idea is to formulate the problem as bilevel programming by minimizing the bias-variance term as the upper-level objective with a lower-level differential privacy machine learning model. The self-adaptive clipping threshold parameters are calculated by balancing the fairness bias-variance and per-class accuracy terms simultaneously. Our contributions can be summarized as follows:

- FairDP uses a self-adaptive clipping threshold to adjust the instance influence in each class, so the model accuracy for each class is calibrated based on their privacy cost through fairness balancing. The class utility reduction is semblable for each class in FairDP.

- To our knowledge, we are the first to introduce bilevel programming to private learning, aiming to mitigate the disparate impact on model accuracy. We further design an alternating scheme to learn the self-adaptive clipping and private model simultaneously.

- Our experimental evaluation shows that FairDP strikes a balance among privacy, fairness, and accuracy by performing stratified clipping over different subclasses.

The following is the road-map of this paper. Section 2 describes the proposed FairDP algorithm. In Section 3, we provide a brief but complete introduction to related works in privacy-aware learning, fairness-aware learning, and the intersection of differential privacy and fairness. Extensive experiments are further presented in Section 4, and we finally conclude this paper and discuss some future work in Section 5.

## 2 FairDP: Fair Differential Privacy

### 2.1 The Bilevel FairDP Formulation

Our approach's intuition is to fairly balance the level of privacy (based on the clipping threshold) for each class based on their bias-variance terms, which are introduced by associated DP. The bias-variance terms arise from capping instance influences to reduce the sensitivity of a machine learning algorithm. In detail, a self-adaptive DP mechanism is designed to balance the bias-variance difference among all groups, while the obtained DP mechanism must adapt to the original machine learning problem simultaneously. Recall the definition of the machine learning problem, we assume there are $\ell$ classes and according to the bias-variance term (4) for class $k \in \{1, \cdots, \ell\}$ can be denoted as

$$\tau_k(C_k, \sigma; \theta^*) := 2(1 + \frac{1}{\mu^t L}) \|\nabla f(\theta^*)\| \cdot \eta_k + \eta_k{}^2 + \frac{|\mathcal{G}_k|^2}{n^2} \cdot \sigma^2 C_k{}^2 |\mathbf{1}|, \tag{5}$$

where $C_k$ denotes the clipping parameter for class $k$; $\mathcal{G}_k$ denotes the data sample set for class $k$. As motivated by Section 1, we aim to minimize the associated bias-variance term to obtain a unified clipping parameter for the machine learning problem. However, to mitigate the disparate impact on model accuracy for different classes, we minimize the summation of per-class bias-variance terms. This objective can lead to the self-adaptive clipping threshold among different classes, while the inconsistent DP schemes for different classes should work on the privacy protection on the machine learning model. The self-adaptive clipping threshold parameters should be utilized to learn the original machine learning privately with the DP mechanism. A simple bilevel programming problem[2] (Dempe et al., 2019; Liu et al., 2019) is introduced to model these two goals which influence each

---

[1]Note that we do not attempt to optimize the bias-variance bound in a differentially private way, and we are most interested in understanding the forces at play.

[2]The simple bilevel programming is not to say that the bilevel problem is simple, and it denotes a specific bilevel programming problem.

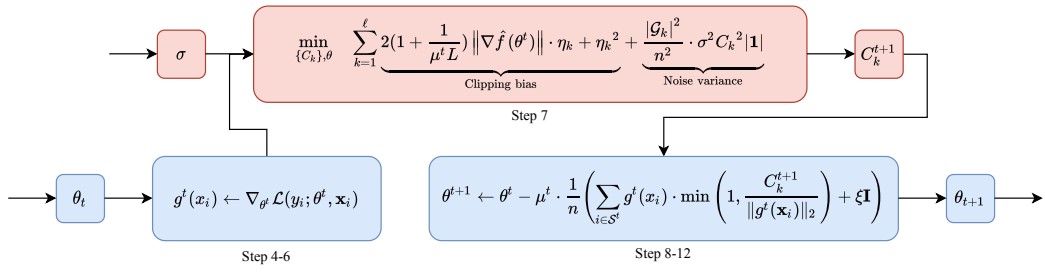

Figure 2: Main flowchart of the FairDP Method (Step 4-12 in Algorithm 1)

other. The formulation can be denoted as follows, i.e.,

$$\min_{\{C_k\},\theta} \quad \sum_{k=1}^{\ell} \tau_k(C_k, \sigma; \theta), \tag{6a}$$

$$\text{s.t.} \quad \theta \in \arg\min_{\theta} \mathcal{L}(\theta; \{\mathcal{G}_k\}_{k=1}^{\ell}), \tag{6b}$$

where the upper-level problem (6a) aims to fairly adjust the clipping threshold parameters for all classes, which is related to the classification model $\theta$; as for the lower-level problem (6b), we aim to learn the classification model based on the differential privacy schema with the self-adaptive clipping threshold $\{C_k\}$. These two objectives are coupled together, although the model of the lower-level problem is determined only by $\theta$. The effect of clipping is reflected through the DP calculation procedure. Guided by the bias-variance term in (6a), the parameters of the DP learning can be finely updated simultaneously with the learning process of the classifiers in (6b).

---

**Algorithm 1:** The FairDP Method

**Input** : Instances $\{(\boldsymbol{x}_1, y_1), \cdots, (\boldsymbol{x}_N, y_N)\}$, objective function $\mathcal{L}(\theta; \boldsymbol{x}, y)$, learning rate $\mu^t$;

1 **Initialize** $\theta_0$;
2 **for** $t \leftarrow 1$ **to** $T$ **do**
3      Randomly sample a batch of instances $\mathcal{S}^t$ with probability $\frac{|\mathcal{S}^t|}{N}$;
     **Compute gradient**
4      **for** $\boldsymbol{x}_i \in \mathcal{S}^t$ **do**
5         Compute $g^t(x_i) \leftarrow \nabla_{\theta^t} \mathcal{L}(y_i; \theta^t, \boldsymbol{x}_i)$;
6      **end**
     **Minimize bias-variance**
7      $C_k^{t+1} \leftarrow \arg\min_{C_k} \tau_k(C_k, \sigma; \theta^t)$;
8      **for** $\boldsymbol{x}_i \in \mathcal{S}^t$ and $y_i = k$ **do**
        **Clip gradient**
9         $\bar{g}^t(x_i) \leftarrow \dfrac{g^t(x_i)}{\max\left(1, \frac{\|g^t(\boldsymbol{x}_i)\|_2}{C_k^{t+1}}\right)}$;
10      **end**
     **Add noise**
11      $\tilde{g}^t \leftarrow \frac{1}{n}\left(\sum_i \bar{g}^t(x_i) + \xi\mathbf{I}\right)$;
     **Noise Gradient Descent**
12      $\theta^{t+1} \leftarrow \theta^t - \mu^t \tilde{g}^t$;
13 **end**
**Output:** $\theta_T$, accumulated privacy cost $(\epsilon, \delta)$.

---

## 2.2 THE FAIRDP METHOD

Calculating the optimal $\theta^*$ and $\{C_k^*\}$ require two nested loops of optimization, and we adopt an alternating strategy to update $\theta$ and $\{C_k\}$ respectively. The FairDP method is summarized in Algorithm 1, while Figure 2 illustrates its main process (i.e., Steps 4-12 in Algorithm 1). For the self-adaptive clipping step (6a) in upper-level, given the current obtained model parameter $\theta^t$, we

can calculate optimal $\{C_k\}$ directly by solving a quadratic programming problem. Based on a batch of training samples $\mathcal{S}^t$ (Step 3), the overall gradient $\nabla f(\theta^t)$ are approximately estimated by $\nabla \hat{f}(\theta^t)$ with the samples in $\mathcal{S}^t$. As for private training step (6b) in lower-level, the updating equation of the classifier parameter can be formulated by moving the current $\theta^t$ along the descent direction of the objective loss in (6b) on a batch training data (Step 4-6):

$$g^t(\boldsymbol{x}_i) := \nabla_{\theta^t} \mathcal{L}(y_i; \theta^t, \boldsymbol{x}_i) \tag{7}$$

After receiving the classifier parameter updating $g^t(\boldsymbol{x}_i)$ from (7), the updated $\theta^{t+1}$ is employed to ameliorate the parameter $\theta$ of the classifier in a DP way as (2) with the obtained clipping parameters $\{C_k\}$ from Step 7 (Step 8-12). All the details can be found in Algorithm 1.

## 3 RELATED WORK

### 3.1 PRIVACY-AWARE LEARNING

Existing literature in differentially private machine learning can be divided into three main categories, input perturbation, output perturbation, and inner perturbation. The input perturbation approach adds noise to the input data based on the differential privacy model. The output perturbation approach adds noise to the model after the training procedure finishes, i.e., without modifying the training algorithm. The inner perturbation approach modifies the learning algorithm such that the noise is injected during training. (Chaudhuri et al., 2011) modifies the objective of the training procedure. DPSGD (Abadi et al., 2016) adds noise to the gradient of each step of the training without modifying the objective.

Limiting users to small influences keeps noise level low at the cost of introducing bias. Several works study how to adaptively bound users' influences and clip the model parameters to improve learning accuracy and robustness. DPSGD fails to provide a detailed analysis of how to choose the gradient norm's truncation level, instead suggesting using the median of observed gradients. Using the median (or any fixed quantile independent of the privacy parameter $\epsilon$) as a cap can yield suboptimal estimations of a sum (Amin et al., 2019). DP-FEDAVG (McMahan et al., 2018) proposes per-layer clipping, which chooses to distribute the clipping budget across layers. (van der Veen et al., 2018) proposes a gradient-aware clipping schedule, which uses a constant factor times the mean private norm of the previous batch as the norm bound for the current batch. (McMahan & Andrew, 2018) does a pre-processing step via the scaling operation. DP-GAN (Zhang et al., 2018) assumes that we have access to a small amount of public data, which is used to monitor the change of gradient magnitudes and set the clipping parameters based on the average magnitudes. (Amin et al., 2019) characterizes the trade-off between bias and variance, and shows that a proper bound can be found depending on properties of the dataset. It does not matter how large or small the gradients are above or below the cutoff, only that a fixed number of values are clipped. (Thakkar et al., 2019) sets an adaptive clipping norm based on a differentially private estimate of a targeted quantile of the distribution of unclipped norms. AdaClip (Pichapati et al., 2019) uses coordinate-wise adaptive clipping of the gradient to achieve the same privacy guarantee with much less added noise.

Previous work either ignores computing the trade-off completely (DPSGD(Abadi et al., 2016) simply uses the empirical median, DP-FedAvg(McMahan et al., 2018) scatter privacy budget evenly over the layers), or requires strong assumptions on the data ((Zhang et al., 2018) assumes the accessibility of public data).

### 3.2 FAIRNESS-AWARE LEARNING

Fairness is a broad topic that has received much attention in the machine learning community. However, the goals often differ from those described in this work. Most researches on fairness-aware machine learning study the *discriminatory prediction problem*: how can we reduce the discrimination against the protected attribute in the predictive decision made by machine learning model (Dwork et al., 2012; Hardt et al., 2016; Kusner et al., 2017). Three common approaches are to preprocess the data to remove information about the protected attribute (Zemel et al., 2013), optimize the objective function under some fairness constraints during training (Zafar et al., 2017), or post-process the model by adjusting the prediction threshold after classifiers are trained (Hardt

et al., 2016). The others study the *discriminatory impact problem* (Kusner et al., 2019): how can we reduce the discrimination arising from the impact of decisions.

In federated learning, AFL (Mohri et al., 2019) has taken a step towards addressing accuracy parity by introducing good-intent fairness. The goal is to ensure that the training procedure does not overfit a model to any one class at another's expense. However, the proposed objective is rigid because it only maximizes the performance of the worst class and has only been applied at small scales (for a handful of devices). q-FFL (Li et al., 2020) reweighs the objective function in FedAvg to assign higher relative weight to classes with higher loss, which reduces the variance of model performance. Although accuracy parity enforces equal error rates among specific classes (Zafar et al., 2017), our goal is not to optimize for identical accuracy across all classes, and we focus on the inequality introduced by differential privacy.

### 3.3 DIFFERENTIAL PRIVACY AND FAIRNESS

Recent works study the connection between achieving privacy protection and fairness. (Dwork et al., 2012) proposes a notion of fairness that is a generalization of differential privacy. ADFC (Ding et al., 2020), DP-POSTPROCESSING/DP-ORACLE-LEARNER (Jagielski et al., 2019), PFLR* (Xu et al., 2019) achieve fairness in addition to enforcing differential privacy in the private model. Most existing work focuses on preventing private information extraction while reaching acceptable fairness performance. Minority work focuses on the accuracy disparity among classes with different protected attributes caused by differential privacy. DPSGF-F (Xu et al., 2020) prevents the disparate impact of the privacy model on model accuracy across different groups by scales the clipping bound with relative ratio. Different from their restriction of the fraction of instances with gradient norms exceeding the clipping threshold, our analyses quantify the bias-variance with sufficient decrease difference between non-private and private learning.

## 4 EXPERIMENTS

This section reports our evaluation of the fair differential private learning on some benchmark datasets from text to vision. We use PyTorch 1.6.0 to implement all the methods with only one NVIDIA GeForce RTX 2080Ti.

**Datasets**: three datasets are used, including the Adult (Dua & Graff, 2017), the Dutch (Kamiran & Calders, 2011) and the Unbalanced MNIST (LeCun et al., 1998)[3]. The details can be found in Appendix B.

**Comparison methods**: 1) SGD: non-private learning without clipping and noise-addition; 2) DPSGD (Abadi et al., 2016): private learning with flat clipping; 3) DP-FedAvg (McMahan et al., 2018): private learning with per-layer clipping; 4) Opt-Q: private learning with $(1 - \sqrt{2/\pi} \cdot \sigma/e)$-quantile clipping, which is adapted from (Amin et al., 2019) and details can be found in Appendix C.1; 5) DPSGD-F (Xu et al., 2020): private learning with clipping proportional to the relative ratio of gradients exceeding the threshold. More details in Appendix C.

**Settings and hyper-parameters**: Without loss of generality, we assume that the function $f$ is 1-Lipschitz. For the Adult and Dutch datasets, we employ a logistic regression machine learning model. Then for logistic regression, the DP-FedAvg will degenerate to classical DPSGD. The noise scale $\sigma$, clipping bound and $\delta$ are set to be $1$, $0.5$ and $10^{-5}$ respectively. For the Unbalanced MNIST dataset, we employ a neural network with 2 convolutional layers and 2 fully-connected layers. The noise scale $\sigma$, clipping bound and $\delta$ are set to be $1$, $1$ and $10^{-5}$ respectively. More Details can be found in Appendix D.3.

To evaluate the efficiency of the proposed FairDP, we aim to complete the following three tasks, i.e., 1) Fairness performance: not only the utility loss is small, but also we can obtain more fair utility loss on different classes; 2) Privacy performance: the proposed FairDP method can preserve the privacy of training data; 3) Adaptive performance: the effect of hyper-parameters on compared private methods.

---

[3]The original MNIST dataset is modified by reducing the number of training samples in Class 8 to 500

Table 1: Utility loss for SGD on total population, well-represented group (Class 2 in Unbalanced MNIST and Male in Adult/Dutch) and underrepresented group(Class 8 in Unbalanced MNIST and Female in Adult/Dutch).

| Dataset | Unbalanced MNIST | | | Adult | | | Dutch | | |
|---|---|---|---|---|---|---|---|---|---|
| Class | Total | Class 2 | Class 8 | Total | Male | Female | Total | Male | Female |
| Train Sample size | 54649 | 5958 | 500 | 30162 | 20380 | 9782 | 48336 | $24201_{\pm 50}$ | $24135_{\pm 50}$ |
| Test Sample size | 10000 | 1032 | 974 | 15060 | 10147 | 4913 | 12084 | $6072_{\pm 50}$ | $6012_{\pm 50}$ |
| SGD | $98.85_{\pm.07}$ | $99.28_{\pm.26}$ | $95.96_{\pm.82}$ | $82.62_{\pm.08}$ | $78.45_{\pm.09}$ | $91.22_{\pm.06}$ | $81.74_{\pm.43}$ | $86.26_{\pm.30}$ | $77.17_{\pm.69}$ |
| DPSGD | $-11.86_{\pm1.09}$ | $-6.73_{\pm.79}$ | $-74.31_{\pm5.58}$ | $-7.10_{\pm.06}$ | $-9.31_{\pm.08}$ | $-2.54_{\pm.05}$ | $-3.82_{\pm.57}$ | $-0.73_{\pm.28}$ | $-6.94_{\pm1.02}$ |
| DP-FedAvg | $-10.85_{\pm.63}$ | $-5.95_{\pm.54}$ | $-73.87_{\pm3.69}$ | $-7.10_{\pm.06}$ | $-9.31_{\pm.08}$ | $-2.54_{\pm.05}$ | $-3.82_{\pm.57}$ | $-0.73_{\pm.28}$ | $-6.94_{\pm1.02}$ |
| Opt-Q | $-2.92_{\pm.16}$ | $-3.67_{\pm.38}$ | $\mathbf{+0.13}_{\pm.86}$ | $-0.83_{\pm.10}$ | $-0.61_{\pm.12}$ | $-1.28_{\pm.15}$ | $-0.80_{\pm.20}$ | $+0.47_{\pm.25}$ | $-2.07_{\pm.44}$ |
| DPSGD-F | $-4.38_{\pm.63}$ | $-4.53_{\pm.69}$ | $-16.43_{\pm3.21}$ | $-0.68_{\pm.07}$ | $-0.67_{\pm.09}$ | $-0.71_{\pm.08}$ | $-1.13_{\pm.18}$ | $\mathbf{+0.67}_{\pm.23}$ | $-2.94_{\pm.45}$ |
| **FairDP** | $\mathbf{-0.65}_{\pm.13}$ | $\mathbf{-1.20}_{\pm.69}$ | $-0.55_{\pm1.20}$ | $\mathbf{+0.03}_{\pm.09}$ | $\mathbf{+0.02}_{\pm.13}$ | $\mathbf{+0.05}_{\pm.04}$ | $\mathbf{-0.03}_{\pm.15}$ | $+0.01_{\pm.34}$ | $\mathbf{-0.07}_{\pm.51}$ |

Table 2: Model fairness comparison

| Dataset | Metrics | SGD | DPSGD | DP-FedAvg | Opt-Q | DPSGD-F | **FairDP** |
|---|---|---|---|---|---|---|---|
| Unbalanced MNIST | Atkinson Index | $0.0006_{\pm0.0003}$ | $0.6567_{\pm0.1644}$ | $0.6382_{\pm0.1171}$ | $0.0007_{\pm0.0002}$ | $0.0167_{\pm0.0055}$ | $\mathbf{0.0007}_{\pm0.0004}$ |
| | Gini Index | $0.0474_{\pm0.0087}$ | $0.8791_{\pm0.0724}$ | $0.8524_{\pm0.0463}$ | $0.0643_{\pm0.0092}$ | $0.2317_{\pm0.0326}$ | $\mathbf{0.0581}_{\pm0.0114}$ |
| | MLD | $0.0006_{\pm0.0003}$ | $0.6807_{\pm0.1762}$ | $0.6602_{\pm0.1253}$ | $0.0007_{\pm0.0002}$ | $0.0167_{\pm0.0055}$ | $\mathbf{0.0007}_{\pm0.0004}$ |
| | Theil Index | $0.0006_{\pm0.0003}$ | $0.4346_{\pm0.0826}$ | $0.4267_{\pm0.0588}$ | $0.0007_{\pm0.0002}$ | $0.0159_{\pm0.0051}$ | $\mathbf{0.0007}_{\pm0.0004}$ |
| Adult | Atkinson Index | $0.0253_{\pm0.0002}$ | $0.0695_{\pm0.0001}$ | $0.0695_{\pm0.0001}$ | $\mathbf{0.0232}_{\pm0.0007}$ | $0.0255_{\pm0.0005}$ | $0.0253_{\pm0.0005}$ |
| | Gini Index | $0.3389_{\pm0.0017}$ | $0.5678_{\pm0.0005}$ | $0.5678_{\pm0.0005}$ | $\mathbf{0.3244}_{\pm0.0050}$ | $0.3407_{\pm0.0036}$ | $0.3394_{\pm0.0032}$ |
| | MLD | $0.0253_{\pm0.0002}$ | $0.0697_{\pm0.0001}$ | $0.0697_{\pm0.0001}$ | $\mathbf{0.0232}_{\pm0.0007}$ | $0.0256_{\pm0.0005}$ | $0.0254_{\pm0.0005}$ |
| | Theil Index | $0.0257_{\pm0.0002}$ | $0.0714_{\pm0.0001}$ | $0.0714_{\pm0.0001}$ | $\mathbf{0.0236}_{\pm0.0007}$ | $0.0260_{\pm0.0005}$ | $0.0258_{\pm0.0005}$ |
| Dutch | Atkinson Index | $0.0150_{\pm0.0029}$ | $0.0477_{\pm0.0096}$ | $0.0477_{\pm0.0096}$ | $0.0251_{\pm0.0027}$ | $0.0304_{\pm0.0035}$ | $\mathbf{0.0152}_{\pm0.0022}$ |
| | Gini Index | $0.2726_{\pm0.0274}$ | $0.4855_{\pm0.0486}$ | $0.4855_{\pm0.0486}$ | $0.3536_{\pm0.0189}$ | $0.3886_{\pm0.0222}$ | $\mathbf{0.2750}_{\pm0.0198}$ |
| | MLD | $0.0150_{\pm0.0029}$ | $0.0478_{\pm0.0096}$ | $0.0478_{\pm0.0096}$ | $0.0251_{\pm0.0027}$ | $0.0304_{\pm0.0035}$ | $\mathbf{0.0152}_{\pm0.0022}$ |
| | Theil Index | $0.0150_{\pm0.0029}$ | $0.0477_{\pm0.0096}$ | $0.0477_{\pm0.0096}$ | $0.0251_{\pm0.0027}$ | $0.0303_{\pm0.0035}$ | $\mathbf{0.0152}_{\pm0.0022}$ |

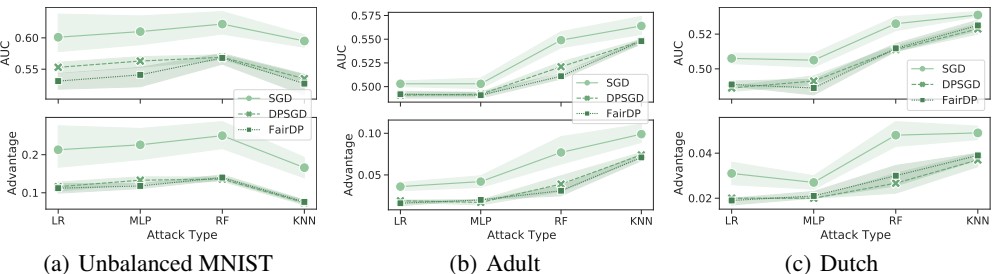

(a) Unbalanced MNIST  (b) Adult  (c) Dutch

Figure 3: Privacy vulnerability comparison

### 4.1 FAIRNESS PERFORMANCE

Table 1 and Table 2 provides the comparison results on model accuracy and fairness after implementing different DP methods. Table 1 presents the utility loss of different private learning methods w.r.t. classical SGD. Table 2 presents the comparison on four fairness indexes (Bureau, 2016), including *Atkinson Index*, *Gini Index*, *MLD* and *Theil Index* (Appendix D.4). In most cases, FairDP has the least accuracy loss from SGD than other methods and offers equal fair statistics as SGD (a lower value is better). Although Opt-Q has little improvement in fairness on the Adult dataset, it reduces both per-class and overall accuracies. Overall, FairDP can outperform other private learning methods on both model fairness and accuracy and balance model fairness and accuracy.

### 4.2 PRIVACY PERFORMANCE

Figure 3 shows the empirical tests for measuring potential memorization from the training data. Same as (Song & Marn, 2020), the attackers can use four classifiers, including Logistic Regression (LR), Multi-Layer Perception (MLP), Random Forest (RF), and K-Nearest Neighbors (KNN). The vulnerability score (Song & Marn, 2020) is set to be the Area Under the ROC-Curve (AUC) and $\max |r_{\mathrm{fp}} - r_{\mathrm{tp}}|$ (Advantage)[4], and lower value means more private. Although FairDP employs self-adaptive clipping parameters, it still maintains a similar level of privacy protection as DPSGD.

---

[4] $r_{\mathrm{fp}}$ and $r_{\mathrm{tp}}$ denotes the false positive rate and true positive rate respectively.

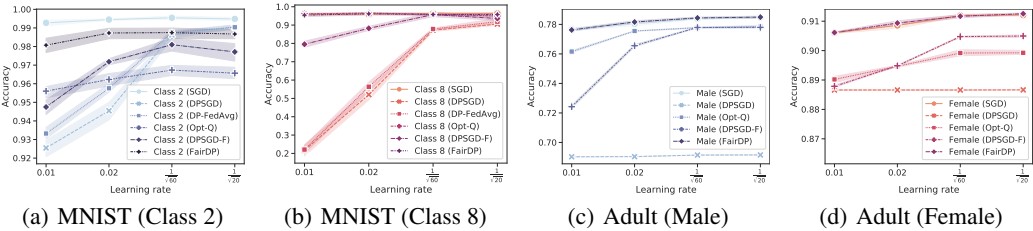

Figure 4: Effect of learning rate on training procedure

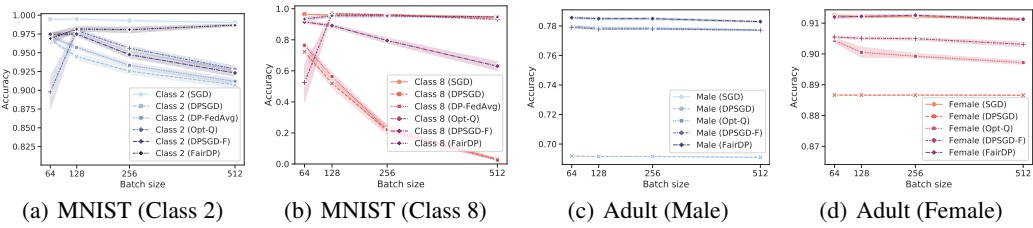

Figure 5: Effect of batch size on training procedure

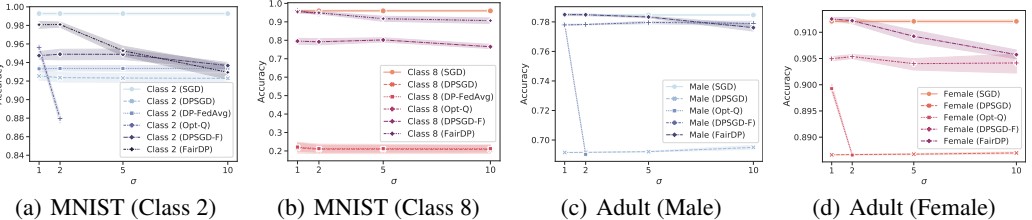

Figure 6: Effect of noise on training procedure

### 4.3 ADAPTIVE PERFORMANCE

(McMahan et al., 2018) claimed that the proper choice for the clipping parameters might depend on the learning rate. If the learning rate changes, the clipping parameter also needs to be re-evaluated. In this experiment, we consider not only the learning rate (Figure 4) but the batch size (Figure 5) and noise (Figure 6), to show the sensitivity of compared private methods on these three hyper-parameters. Figure 4,5 and 6 show that FairDP is insensitive to variations in the hyper-parameters. Because of the space limitation, more results on the Dutch dataset are moved to Appendix E.

## 5 CONCLUSION

Gradient clipping and noise addition, which are the core techniques in DPSGD, disproportionately affect underrepresented and complex classes. As a consequence, the accuracy of a model trained using DPSGD tends to decrease more on these classes vs. the original, non-private model. If the original model is unfair because its accuracy is not the same across all subgroups, DPSGD may exacerbate this unfairness. We propose FairDP, which aims to remove the potential disparate impact of differential privacy mechanisms on the protected group. FairDP adjusts the influence of samples in a group depending on the group clipping bias such that differential privacy has no disparate impact on group utility. In future work, we can further improve our adaptive clipping method from group-wise adaptive clipping to an element-wise adaptive clipping from the user and/or parameter perspectives, and then the model could be fair even to the unseen minority class.

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
