# OpenReview forum: "Fair Differential Privacy Can Mitigate the Disparate Impact on Model Accuracy"
_ICLR.cc/2021/Conference — Reject_

### Official Review · AnonReviewer3 · 2020-10-27
**A fair, but not clearly DP, algorithm.**

**Rating:** 4
**Confidence:** 3

**Review:**

==== Summary of the problems considered and paper contribution

This paper studies an important problem: that differentially private algorithms can have disparate impact on model accuracy for different sub communities. This is an important problem because minority populations often suffer the worst decrease in model accuracy. This paper attempts to solve this problem for supervised learning by introducing an adaptive algorithm version of SGD that attempts to equalize the “bias-variance” trade-off at each iteration. Through experiments they show that their algorithm does indeed result in an improvement in fairness, according to a variety of fairness metrics.

==== Comments

My main concern with the paper is that the privacy claims of Algorithm 1 are not clearly discussed. From my reading, the authors never state whether it is differentially private or not, and this definitely needs clarification. My understanding is that it is NOT differentially private, since the clipping bound is data dependent (according to equation 5), and essentially released in the clear. It’s difficult to tell because I couldn’t find a description of zeta, the amount of noise being added to maintain privacy. If the claim isn’t that Algorithm 1 is DP, then the privacy guarantee is restricted to the results in Figure 3, that attack algorithms perform similarly well on DPSGD, FairDP, and significantly better on SGD (non-private). This is certainly nice to see, but it makes a direct comparison to prior work (that is DP) difficult, and a little unfair.

The experiments are well designed and nicely show how FairDP improves fairness. I am not a fairness expert, so I’m not sure how representative the fairness metrics used in Table 2 are, but it does seem like FairDP improves on prior work in this regard.

==== Presentation

The presentation could use some work. There are many grammatical errors, and occasionally sentences that I couldn’t make sense of (“The self-adaptive threshold parameters should be utilized to learn the original machine learning privately with the DP mechanism”). Mathematical sections are often hard to follow, e.g. algorithm 1 is not self contained, what are N and zeta? I didn’t understand what Figure 1 was trying to show?

The first sentence of section 3.1 seems like a bit dismissive, there is certainly more to the DP ML literature than those three techniques.

---

> ### Author Response · Authors · 2020-11-24
> **Response to Reviewer \#3**
>
> We thank the reviewer for your careful review of our paper.
>
> Please refer to [Privacy Guarantee] in our response to all reviewers.
>
> [Presentation]: Thank you for pointing out the hard-to-understand sentence.
> $N$ is the size of the training dataset. $\xi$ is noise drawn from Gaussian distribution $\mathcal{N}(0, \sigma^2 C^2)$.
> Figure 1(a) shows that clipping is primarily responsible for the disparate impact. Figure 1(b) shows that the smaller $\tau(C, \sigma; \theta^t)$ is, the less the disparate impact. Thus, minimizing $\tau(C, \sigma; \theta^t)$ is a reasonable choice for the fairness objective in our scenario. We revise these parts in our work.

---

### Official Review · AnonReviewer1 · 2020-10-28
**This paper addresses the problem of unbalanced dataset using a differentially private SGD method called FairDP.**

**Rating:** 4
**Confidence:** 4

**Review:**

This paper addresses the problem of an unbalanced data set. In particular, the accuracy on the well-represented classes is higher than the accuracy in underrepresented classes in an unbalanced dataset. This paper shows that DPSGD makes the problem of an unbalanced dataset even worse and decreases accuracy on the underrepresented class significantly. Further, this paper introduces a modification of DPSGD, which can increase the underrepresented class's accuracy.

Strengths:
1. This paper a self-adaptive DP mechanism to address the problem of an unbalanced dataset.
2. Extensive numerical examples that demonstrate the performance of FairDP.

Weaknesses:
1. This paper situates itself in Fairness literature. In the fairness literature, we should have a dataset that includes sensitive attributes and equalize a fairness measure across different (demographic groups). This paper addresses the problem of an unbalanced dataset and tries to equalize the accuracy for different classes (labels).
2. This paper modifies the DPSGD method without providing that the modified method can achieve differential privacy. In order to make sure the FairDP is $(\epsilon,\delta)$-differentially private, they have to theoretically find the privacy parameters with respect to the training datasets. DPSGD (Abadi et al. (2016)) uses the Gaussian mechanism and calculates privacy cost using differential privacy definition. In the current paper, we do not see any privacy analysis, and we are not sure whether FairDP satisfies the differential privacy definition. If FairDP does not satisfy the DP definition or has a very large privacy loss compared to DPSGD, it is not fair to compare FairDP with DPSGD.
3. This paper compares its own method with other methods that their goal is not to address the problem of an unbalanced dataset. In order to make the experiment more informative, I suggest authors compared FairDP with other algorithms that have been designed for addressing unbalanced datasets.

---

> ### Author Response · Authors · 2020-11-24
> **Response to Reviewer \#1**
>
> Thank you for your insightful comments.
>
> Q1: [Fairness Literature Review]
>
> A1: We agree that a standard definition of accuracy parity is to enforce equal precision among the protected groups. However, our objective is to encourage the uniformity of the utility loss among groups partitioned by labels, conceptually similar to [AFL] and [q-FedAvg].
>
> [AFL]: M. Mohri, G. Sivek and A.~T. Suresh, Agnostic Federated Learning. ICML, 2019.
>
> [q-FedAvg]: T. Li, M. Sanjabi, A. Beirami and V. Smith, Fair Resource Allocation in Federated Learning. ICLR, 2020.
>
> Q2 \& Q3:
> Thanks for your comments. Please refer to [Privacy Guarantee] and [Class Imbalance Problem] in our response to all reviewers.

---

### Official Review · AnonReviewer4 · 2020-10-29
**Reject**

**Rating:** 4
**Confidence:** 4

**Review:**

The paper introduces an algorithm for mitigating disparate impact of private learning (DP-SGD) on different groups of a given population. In each iteration of DP-SGD, instead of using a uniform gradient clipping threshold for all groups, the proposed Fair DP-SGD algorithm uses an optimal clipping threshold (one that minimizes the bias-variance tradeoff) for each group separately. The authors include experimental results to show how well their algorithm performs compared to state-of-the-art algorithms.

While the problem considered by the authors is very interesting and has impact on real world, I recommend rejection. The major concern I have with this work is that it lacks a formal (differential) privacy statement. I am not even entirely sure that the proposed algorithm is actually differentially private because the step that finds the optimal clipping thresholds seems to use the non-noisy mini-batch gradients without any privatization (please clarify if my understanding is not correct). In any case, if the proposed algorithm is claimed to be (epsilon, delta)-DP then there must be a rigorous proof for it. Also, in the experiments I don't see any reported values for epsilon? Are different methods compared with the same value of epsilon?

Other comments:

-In addition to a formal privacy statement, the authors should formally define the notions of "privacy" and "fairness" that they use in the paper. Overall, I believe this work can have a better formalization.

-As mentioned earlier, I cannot find the values of epsilon in the experiments. The authors could for e.g. use moments accountant to find the total privacy loss in their experiments.

-When the model is logistic regression (which is the adopted model for 2/3 of the datasets in the experiments) and if the input data is normalized, then the Lipschitz constant L of the (logistic) loss function is a small constant. So in this case clipping the gradients is not necessary because the norm of gradients is always bounded by the Lipschitz constant L which is small and the added noise can be calibrated to L. I think in the case of logistic regression, the authors should also compare their method with a private SGD algorithm that simply adds noise with scale ~ L without any clipping.

-I'm not sure if I understand the optimization problem given in 6a, 6b and how the algorithm is solving it. In particular, the constraint set of the problem seems to be all models with optimal risk (absent any fairness, privacy). But are you actually solving this problem? I.e., does the model output by the algorithm fall into this constraint set?

-The gradients are sometimes denoted by g^t := \nabla L and other times by \nabla f (see for e.g. section 2.2). Is f the same as the loss function L? It would be better if a consistent notation was picked for gradients.

---

> ### Author Response · Authors · 2020-11-24
> **Response to Reviewer \#4**
>
> We greatly appreciate the reviewer’s detailed review and suggestions to improve the paper.
>
> Please refer to [Privacy Guarantee] in our response to all reviewers.
>
> [Lipschitz Constant of Logistic Regression]: Thank you for providing valuable information about the Lipschitz constant of the logistic regression. We will provide experiment results in our next version.
>
> [Optimization Problem]: Thank you for this comment. We will conduct more experiments to explore whether the solution falls into the constraints.
>
> [Notation Consistency]: You are right about the notation. The gradients should be denoted by $\nabla f$. We will check and update our notations throughout the paper carefully.

---

### Official Review · AnonReviewer2 · 2020-10-29
**The paper proposes a fair + differentially private sgd procedure for learning classification models on unbalanced, sensitive datasets.  Their key idea is to use different clipping thresholds for individuals in different classes, treating this as a knob that can be used to control the influence of individual records to the gradient.**

**Rating:** 5
**Confidence:** 3

**Review:**

Strong Points:

1. Fairness and privacy are both relevant research areas, and knowing how they interact and finding ways to achieve both is certainly important.
2. The proposed approach makes sense at a high level, and seems to succeed in achieving the goal.

Weak Points:

1. Possible problem with DP-SGD (see below).
2. Some missing details in the algorithm (see below).
3. Experiments are somewhat confusing (see below).

Other Notes:

The weak points listed above are more like questions that could use further clarification.

1. In Equation (2) you describe DP-SGD as adding scalar noise * vector of ones (same noise value to each entry).  However, this is not how DP-SGD usually works.  Is this a typo or can you clarify why you are adding noise in this way?
How are you calculating the total privacy cost (eps, delta) at the end of the algorithm.  Do different groups get different levels of privacy protection, and if so what does that mean, and is that considered a fairness violation as well?


2. In related work you describe an approach to fairness that works by simply reweighting the loss function to boost impact of under-represented classes.  Wouldn’t that be a simpler approach than what you are doing but provide the same benefits?  Should potentially be considered as a baseline.

3.
a) Fig 3 doesn’t really make sense as a line plot to me.
b) Also measuring privacy by strength of adversaries does not seem natural here, I would prefer to measure privacy using the parameters eps/delta.
b) Why is DPSGD doing so poorly on the total loss?  I have to wonder if it is just not well-tuned?  My expectation is that mechanisms that don’t impose fairness requirements would achieve better total loss, although more uneven between the classes.  Results differing from this are surprising, and warrant further discussion/explanation.

---

> ### Author Response · Authors · 2020-11-24
> **Response to Reviewer \#2**
>
> We thank the reviewers for their meaningful and valuable comments, which help to improve the quality of our work.
>
> Q1: [Problem with DPSGD]
> Thank you for pointing out the typo. We intend to describe DPSGD as adding a vector with the same shape as $\theta$, with each entry a noise value drawn independently from Gaussian distribution. We revise Equation (2) as $\mathcal{N}(0, \sigma^2 C^2 \mathbf{1})$.
>
> Please see [Privacy Guarantee] in our response to all reviewers.
>
> We don't think that different clipping thresholds for different groups means a fairness violation. Fairness is not only about offering clipping thresholds in matching the same amounts (DPSGD), but also focuses more on providing clipping thresholds proportionally to achieve a fair outcome for the groups. Please let us know if this does not answer your question.
>
> Q2:
> Please see [Class Imbalance Problem] in response to all reviewers.
>
> Q3: [Experiments]
> a) Thank you for this comment. We revise Figure 3 to a bar plot.
> b) You are correct that we have slight differences in accuracy relative to prior work (DPSGD-F). Their experimental results situate in the range of our reported mean and variance.
>
> [DPSGD-F] D. Xu, W. Du, X. Wu: Removing Disparate Impact of Differentially Private Stochastic Gradient Descent on Model Accuracy. 2020.

---

### Author Response · Authors · 2020-11-24
**Responses to All Reviewers**

We thank all reviewers for their helpful comments. We first address shared concerns and then respond to specific comments below.

[Privacy Guarantee]: We are working on calculating the total privacy parameters ($\epsilon$, $\delta$) and theoretically proving that FairDP is differentially private. We cannot provide the re-worked privacy statement of FairDP in time. Thank you for the reviewers' interests and questions. We will update the proof after finishing our re-work on FairDP.

[Class Imbalance Problem]: Thank you for your suggestion. Addressing the class imbalance problem is a significant problem but beyond the scope of our paper. As declared in Section 3.2, ``Our goal is not to optimize for identical accuracy across all classes, and we focus on the inequality introduced by differential privacy.''

---

### Decision · Program_Chairs · 2021-01-07
**Final Decision**

**Decision:**

Reject

**Comment:**

This paper proposes an algorithm to address the disparate effect that DP has on the accuracy of minority/low-frequency sub-populations. Unfortunately the work does not actually guarantee or analyze the resulting privacy guarantees. In particular it may provide much worse privacy (or no privacy at all) to the minority subpopulation.
The paper also calls their algorithm "fair" without using an accepted term or a careful discussion of what an algorithm needs to satisfy to be considered "fair". Using a more technical term such "reducing the accuracy disparity" would make much more sense.